# Cross-Dataset Sensor Alignment: Making Visual 3D Object Detector Generalizable

**Liangtao Zheng**[1,5*]    **Yicheng Liu**[2,1]    **Yue Wang**[3]    **Hang Zhao**[2,1,4†]

[1]Shanghai Qi Zhi Institute   [2]IIIS, Tsinghua University
[3]University of Southern California   [4]Shanghai AI Lab  [5]UC San Diego

**Abstract:** While camera-based 3D object detection has evolved rapidly, these models are susceptible to overfitting to specific sensor setups. For example, in autonomous driving, most datasets are collected using a single sensor configuration. This paper evaluates the generalization capability of camera-based 3D object detectors, including adapting detectors from one dataset to another and training detectors with multiple datasets. We observe that merely aggregating datasets yields drastic performance drops, contrary to the expected improvements associated with increased training data. To close the gap, we introduce an efficient technique for aligning disparate sensor configurations —a combination of camera intrinsic synchronization, camera extrinsic correction, and ego frame alignment, which collectively enhance cross-dataset performance remarkably. Compared with single dataset baselines, we achieve 42.3 mAP improvement on KITTI, 23.2 mAP improvement on Lyft, 18.5 mAP improvement on nuScenes, 17.3 mAP improvement on KITTI-360, 8.4 mAP improvement on Argoverse2 and 3.9 mAP improvement on Waymo. We hope this comprehensive study can facilitate research on generalizable 3D object detection and associated tasks.

**Keywords:** 3D Object Detection, Model Generalization, Autonomous Driving

## 1 Introduction

3D object detection has emerged as an important task for robots. For example, autonomous vehicles require precise localization of traffic participants, such as cars, pedestrians, and bicycles, to ensure safe driving. Consequently, 3D object detection has garnered significant attention, leading to improved accuracy across several benchmarks [1, 2, 3]. Nonetheless, a common limitation of existing methods [4, 5, 6, 7, 8, 9, 10] is their tendency to be trained and evaluated on the same benchmark.This practice overlooks the influence of data diversity, often under the assumption that training and testing datasets are uniformly distributed - an assumption that might not always stand in real-world applications, especially when a detector is deployed across varied vehicle models. This raises concerns regarding the capability of these methods to learn from and adapt to a diverse range of datasets.

To investigate this, we initiated a straightforward experiment, training a model on one dataset and testing it on another. The results revealed a severe decline in performance: a detector trained on the Argoverse2 [2] dataset experiences a 70.4% performance drop when evaluated on the Waymo [3] dataset, compared to the counterpart trained directly on Waymo. Then, we add the nuScenes [11] dataset to augment the training data volume. However, the model with additional data fails to achieve meaningful performance (5.2 mAP) when tested on Waymo. This outcome amplifies a critical question within the field of 3D perception: how to effectively utilize diverse data sources during training.

---

[*]l9zheng@ucsd.edu
[†]Corresponding at: hangzhao@mail.tsinghua.edu.cn

7th Conference on Robot Learning (CoRL 2023), Atlanta, USA.

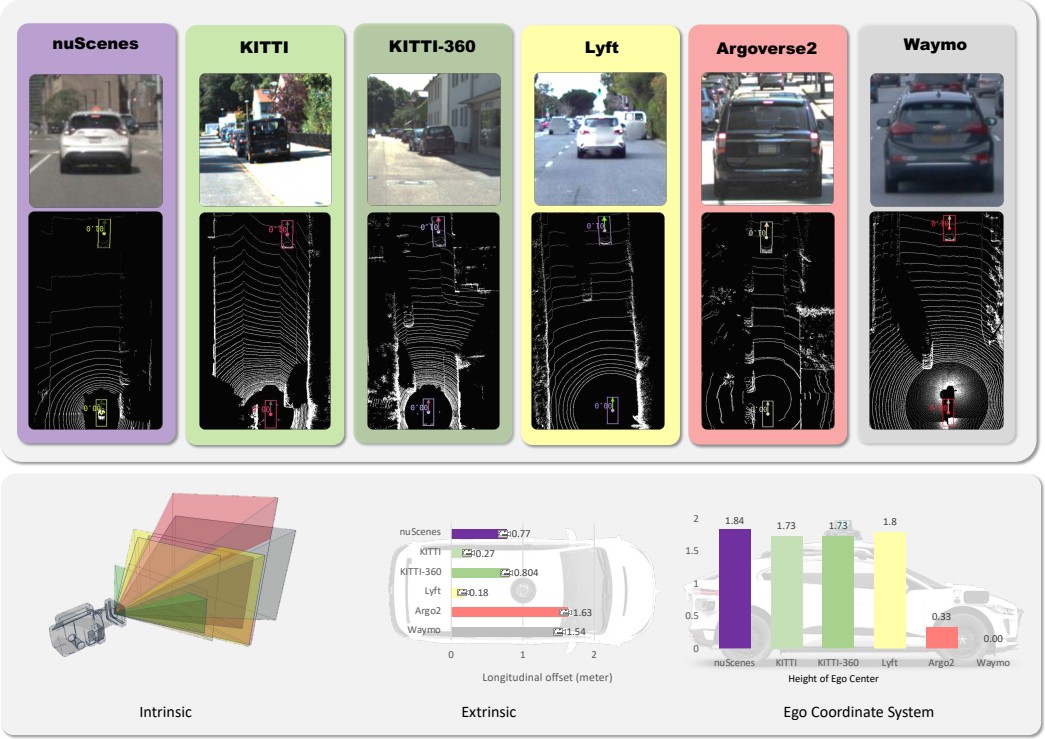

Figure 1: Top: **Similar cars in different datasets.** We provide image patches of the same resolution and LiDAR point clouds of the same scale. Objects in similar 3D shapes and distances differ significantly in 2D shapes. Bottom: **Different Sensor Suite Parameters.** The focal length and resolution vary, forming different imaging planes. The camera longitudinal offset from the ego center also varies.

Why incorporating additional data hampers model performance. What differences between the datasets are responsible for such catastrophic failures? Suspecting the disparities in sensor configurations to be the issue, we conduct another concise experiment to validate this hypothesis. In Fig. 2, we train a 3D detector on Waymo images with a 2070 mm focal length and evaluated it on images of different focal lengths, achieved through digital zooming. Optimal performance was observed when the focal lengths of both training and testing images were aligned. This finding remained consistent even when applied to disparate datasets, *e.g.* Argoverse2.

This observation reinforces our hypothesis concerning the significant influence of camera parameter variations on 3D detection. The underlying reason is rooted in the nature of imaging [12]: an image serves as a 2D projection, capturing and rendering visual information from the 3D physical world. As depicted in Fig. 1, varying sensor configurations lead to unique projections.

We term this issue as *sensor misalignment* across different datasets. Our in-depth analysis underscores the pivotal roles of intrinsic, extrinsic, and ego coordinate system in this misalignment, as detailed in (§ 3.4). To mitigate this issue, we introduce straightforward strategies that leverage sensor parameters to compensate for biases in input signals. First, we resize all the input images to unify the focal lengths. Second, an Extrinsic Aware Feature Sampling is incorporated into the detection pipeline to counteract the effects of camera translations. Third, ego frame alignment is employed to resolve ambiguities in the ego frame definition, addressing the intertwined issues of camera height and ego center. Our method yields a massive improvement in the generalization capability of the detector, with an average increment of 29.5 mAP when adapting nuScenes to other datasets. In summary, the main contributions of this paper include:

- A thorough evaluation pinpointing the critical issues resulting in performance decline during cross-dataset testing and multi-dataset training. Our findings highlight three key elements: intrinsic, extrinsic, and the ego coordinate system.

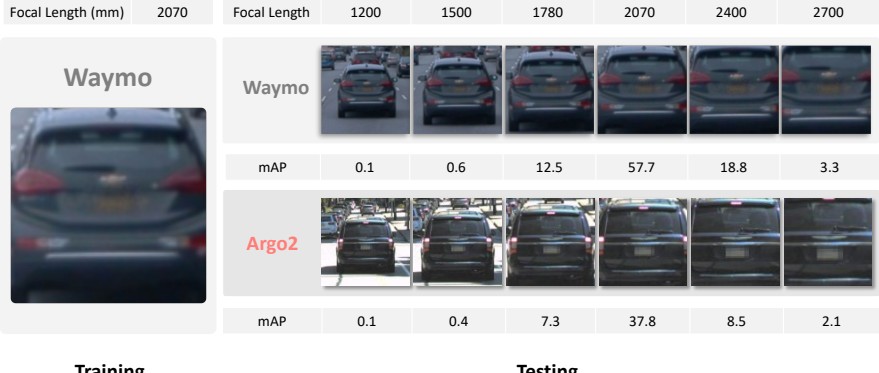

| Focal Length (mm) | 2070 | | Focal Length | 1200 | 1500 | 1780 | 2070 | 2400 | 2700 |
|---|---|---|---|---|---|---|---|---|---|

Figure 2: Training and testing on the same focal length setting gives optimal results.

- A simple yet effective sensor alignment method to counteract this issue by correcting the input signals, leading to notable performance boosts across all evaluated datasets.
- Remarkable performance enhancements across various datasets. Compared to direct transfer, our approach achieves an average improvement of 29.5 mAP in cross-domain adaptability. Additionally, our jointly trained models outperform those trained on individual datasets, even surpassing models specifically trained on Lyft, KITTI, and KITTI-360 datasets without utilizing these datasets during the training phase.

## 2  Related Work

**Camera-based 3D Detection.** Recently, Significant advancements have been made in camera-based 3D object detection in the Bird's Eye View (BEV) space [4, 6, 13, 5, 14, 15, 7, 16, 17]. The majority of these approaches transform 2D image features into 3D space by camera parameters. Inspired by LSS [18], certain methods [6, 7, 16] estimate depths for image features and shoot them to a predefined 3D grid to create a BEV feature map. Another branch of methods utilizes object queries [19]. They generate 3D queries [4, 13, 5, 8] and project them onto the image plane to sample features. Notably, many works are derivatives of DETR3D [4] and BEVDet [6], sharing substantial similarities.

**Domain Adaptation.** This line of methods aims to improve model performance from the source to the target domain [20, 21, 22, 23, 24, 25, 26, 27, 28, 29]. In paper [30], the authors explore the impact of data distribution on the cross-dataset performance of LiDAR-based 3D detectors. In [31], Wang et al. analyze the impact of camera intrinsic parameter on image features and depth estimation based on BEVDepth [7]. Diverging from focusing on a single dataset, our study extends experiments to *training* on multiple datasets, addressing misalignments in distribution during both training and inference phases. Besides intrinsic, we include the role of camera extrinsic and the ego coordinate system in causing such misalignments.

**Multi-dataset training.** A number of studies have improved generalization capability by training on combined datasets. In monocular depth estimation, MiDAS [32] illustrates the efficacy of mixing five datasets from complementary sources. Uni3D [31] focuses on joint training strategies and their impact on LiDAR-based 3D object detection. Both studies mentioned that naively adding datasets does not guarantee improvement. We echo this sentiment in vision-centric 3D object detection, and propose a solution through cross-dataset sensor alignment.

## 3  Experiment and Analysis

### 3.1  Experiment Protocols

**Datasets.** Our experiments involve six datasets: Argoverse2 [2], KITTI [1], KITTI-360 [33], Lyft [34], nuScenes [11] and Waymo [3], with a focus on camera-based 3D object detection data. Each of these datasets offers ground-truth 3D bounding box labels for various object types. An overview of these datasets is available in Table 1. We have standardized different dataset formats to the MMDetection3D [35] format for a cohesive analysis.

Table 1: Datasets overview.

| Dataset | Abbr. | #frame | Image resolution | Object type | #RGB camera |
|---------|-------|--------|------------------|-------------|-------------|
| Argoverse2 | A | 26,687 | (2048, 1550) | 26 | 7 |
| Kitti | K | 7,481 | (1224, 370) | 8 | 2 |
| Kitti-360 | K360 | 61,569 | (1408, 376) | 26 | 4 (2 fisheye) |
| Lyft | L | 22,680 | (1920, 1080)/(1224, 1024) | 9 | 7 |
| nuScenes | N | 28,130 | (1600, 900) | 8 | 6 |
| Waymo | W | 39,614 | (1920, 1280) | 3 | 5 |

**3D Object Detectors.** Aiming for a method applicable to both multi-view and single-view detection, we employ BEV detectors, DETR3D [4], and BEVDet [6] as our baselines, steering clear of image-based detectors like FCOS3D [36] due to their proven limitations in multi-view scenarios [4].

**Metrics.** We adopt the LET-3D-AP [37] metric in line with the 2022 Waymo Open Dataset Challenge [3]. All dataset categories are merged into three primary classes: vehicle, pedestrian, and bicycle. We present the LET-3D-AP for each using IoU thresholds of $0.5$, $0.3$, and $0.3$ within a unified perception range of 51.2 meters.

For clarity, we only showcase monocular detection results of DETR3D, and the average mAP for the three classes in the main paper. More detailed insights, including multi-view detection results, BEVDet experiments, individual class mAP, and training specifics, are elaborated in the Appendix.

### 3.2 Training on one dataset and testing across datasets

We began by training detectors on individual datasets and testing their performance across different datasets. The "Direct" block in Table 2 illustrates both in-domain (*i.e.*, train and test on the same dataset) and cross-domain performance. According to the numbers in bold font, DETR3D exhibits satisfactory in-domain mAP on Waymo and Argoverse2 (AV2) but falters on Lyft, nuScenes, KITTI, and KITTI-360. One cause of the declination is diverse data collection conditions: Lyft is collected by 20 different autonomous vehicles, while nuScenes includes data in Singapore and the USA, causing difficulty in the model's convergence. Another cause is insufficient data volume and pixel limitation. KITTI only has 4,000 training samples, and KITTI-360 has the smallest focal length, rendering pedestrians and cyclists nearly undetectable. Regarding cross-domain mAP, it almost drops to 0 for most dataset pairs. This downfall cannot be attributed to domain shifts in the environment or object size, evidenced by the failed transfer between KITTI and KITTI-360, which are collected in the same city.

An auxiliary experiment depicted in Fig. 2 underscores the model's sensitivity to focal length. The model's performance dips on Waymo itself with focal length deviation but improves markedly when AV2 is resized to Waymo's focal length. This indicates intrinsic variation, as depicted in Fig. 1, to be a core issue. In the ensuing subsection, we demonstrate that merely expanding the volume of training data is insufficient to overcome this challenge.

### 3.3 Training on multiple datasets and testing across datasets

We augment data diversity by sequentially adding datasets into the training mix and develop six distinct models [3]. The "Direct" section of Table 3 showcases fluctuating performance metrics as the dataset expands. There's an mAP increase for nuScenes from 36.3% to 46.2% but an observable decline upon the integration of KITTI-360. Similarly, KITTI's mAP recedes from 41.4% to 36.3%.

From a broader view, neither cross-domain nor in-domain performance (avg.S and avg.T) achieve meaningful improvement despite the increased data volume. The detector continues to overlook the intrinsic disparities within the input images, even with a mixed dataset.

---

[3]Here, our adding order is nuScenes, AV2, Lyft, KITTI, KITTI-360, and Waymo. However, the trend of performance is invariant to the order. See section 5.4 in the Appendix for another result starting with Waymo.

Table 2: Cross-domain performance DETR3D [4] trained on a single dataset. "Direct" means direct transfer. "avg.T" stands for average in target domains. The **bold** font highlights the in-domain performance. See § 4 for details of "K-sync", "E-aware", and "Ego-sync".

| Setting | src\dst | N | A | L | K | K360 | W | avg.T |
|---|---|---|---|---|---|---|---|---|
| Direct | N | **36.3** | 0.8 | 1.8 | 0.0 | 0.0 | 1.1 | 0.7 |
| | A | 0.2 | **48.0** | 0.1 | 0.0 | 0.0 | 17.4 | 3.5 |
| | L | 0.5 | 0.1 | **37.3** | 0.4 | 0.0 | 0.1 | 0.2 |
| | K | 2.8 | 1.2 | 0.0 | **24.5** | 1.1 | 0.7 | 1.2 |
| | K360 | 0.1 | 0.2 | 0.0 | 3.2 | **26.1** | 0.1 | 0.7 |
| | W | 0.1 | 8.9 | 0.0 | 0.0 | 0.0 | **58.8** | 1.8 |
| K-sync | N | **40.8** | 25.5 | 18.6 | 29.7 | 18.0 | 23.4 | 23.0 |
| | A | 13.2 | **51.4** | 7.5 | 6.6 | 4.6 | 38.8 | 14.1 |
| | L | 1.0 | 1.3 | **44.0** | 8.1 | 5.7 | 1.5 | 3.5 |
| | K | 2.4 | 1.2 | 1.2 | **31.0** | 6.1 | 0.5 | 2.3 |
| | K360 | 14.6 | 14.7 | 7.3 | 34.6 | **34.7** | 8.2 | 15.9 |
| | W | 14.5 | 37.8 | 14.3 | 9.4 | 5.6 | **57.7** | 16.3 |
| K-sync, E-aware, and Ego-sync | N | **43.1** | 33.6 | 32.8 | 33.0 | 18.4 | 33.0 | 30.2 |
| | A | 24.4 | **48.1** | 34.1 | 18.1 | 8.7 | 37.4 | 24.5 |
| | L | 15.7 | 19.6 | **47.1** | 20.0 | 12.9 | 18.9 | 17.4 |
| | K | 7.1 | 8.7 | 10.2 | **29.1** | 9.3 | 2.4 | 7.5 |
| | K360 | 13.9 | 17.7 | 16.6 | 39.1 | **36.7** | 8.4 | 19.1 |
| | W | 25.4 | 38.2 | 33.6 | 21.2 | 11.7 | **57.6** | 26.0 |

## 3.4 Analysis

In this section, we model 3D-2D correspondence of objects and scrutinize 3D detection pipeline. Our analysis reveals that apart from intrinsic, extrinsic and ego coordinate system also influence detection performance.

**3D-2D correspondence of objects.** We begin by examining the projection of an 3D object via pinhole camera model. With a frontal camera of focal length $f_x$ at coordinates $(t_x, t_y, t_z)$ relative to the ego frame origin and an object at $(x, y, z)$, having a 3D size $S$ and pixel size $s_{pixel}$, their relationship can be formulated as:

$$s_{\text{pixel}} = f_x \times \frac{S}{x - t_x}, \tag{1}$$

where $x - t_x$ indicates the depth in the camera frame, and each variable in the equation is a scalar. Changes in $f_x$ and $t_x$ result in different $s_{pixel}$ values, causing the same object to appear differently —a factor often overlooked in cross-dataset training and testing.

**3D detection pipeline.** To understand the impact of Eq. (1) on 3D detection, we trace the detection process. Initially, the detector projects a 3D query point $\mathbf{p}_0(x_0, y_0, z_0)$ to a 2D coordinate $(u, v)$. It then samples image features and predicts the object's attributes using both positional and semantic information:

$$\mathcal{H} : (I(u - s, v - s, u + s, v + s), \mathbf{p}_0) \longrightarrow \hat{\mathbf{b}}_0, \hat{c}_0, \tag{2}$$

where $I(u - s, v - s, u + s, v + s)$ denotes an image patch centered at $(u, v)$ with dimensions $2s \times 2s$. We simplify our analysis by treating this patch as the image feature, skipping the feature extraction process. The vector $\hat{\mathbf{b}}_0$ denotes the predicted position and size, while the scalar $\hat{c}_0$ is the classification score. Additionally, according to the pinhole camera model, the projection from $\mathbf{p}_0$ to $(u, v)$ follows:

$$d(u, v, 1)^T = \mathbf{KTp}_0, \tag{3}$$

with $\mathbf{K}$ and $\mathbf{T}$ being the intrinsic and extrinsic matrices, and $d = x_0 - t_x$ being the depth of $\mathbf{p}_0$ in the camera frame, embodied in $\mathbf{KTp}_0$. The mapping function $\mathcal{H}$ is learned during training.

Considering the query point is the object center, and the image patch contains the object, the implications of Eq. (1) extend to Eq. (2): variations in intrinsic $\mathbf{K}$ and extrinsic $\mathbf{T}$ scale the object

Table 3: Performance of DETR3D trained on **multiple** datasets. "Direct" means direct merge for training and direct transfer for testing. "avg.T" stands for the average in target domains. "avg.S" stands for the average in source domains. See § 4 for details of "K-sync", "E-aware" and "Ego-sync".

| Setting | src\dst | N | A | L | K | K360 | W | avg.S | avg.T |
|---|---|---|---|---|---|---|---|---|---|
| Direct | N | 36.3 | 0.8 | 1.8 | 0.0 | 0.0 | 1.1 | 36.3 | 0.7 |
| | +A | 40.5 | 49.2 | 0.5 | 0.0 | 0.0 | 5.2 | 44.9 | 1.4 |
| | +L | 41.6 | 50.5 | 43.7 | 0.0 | 0.0 | 3.8 | 45.3 | 1.3 |
| | +K | 41.5 | 49.7 | 46.0 | 41.4 | 1.1 | 3.6 | 44.6 | 2.4 |
| | +K360 | 42.6 | 54.3 | 46.8 | 36.3 | 29.7 | 3.3 | 41.9 | 3.3 |
| | +W | 46.2 | 53.7 | 49.4 | 39.5 | 29.7 | 61.9 | 46.7 | - |
| K-sync | N | 40.8 | 25.5 | 18.6 | 29.7 | 18.0 | 23.4 | 40.8 | 23.0 |
| | +A | 45.5 | 50.0 | 25.1 | 35.8 | 21.3 | 44.2 | 47.8 | 31.6 |
| | +L | 46.8 | 53.2 | 55.1 | 37.8 | 23.1 | 45.3 | 51.7 | 35.4 |
| | +K | 47.4 | 53.5 | 53.6 | 57.8 | 21.8 | 44.4 | 53.1 | 33.1 |
| | +K360 | 50.2 | 54.4 | 54.0 | 60.2 | 39.6 | 44.7 | 51.7 | 44.7 |
| | +W | 51.8 | 55.3 | 56.6 | 61.9 | 40.7 | 63.7 | 55.0 | - |
| K-sync, E-aware, and Ego-sync | N | 43.1 | 33.6 | 32.8 | 33.0 | 18.4 | 33.0 | 43.1 | 30.2 |
| | +A | 52.1 | 52.7 | 38.4 | 42.2 | 23.2 | 40.7 | 52.4 | 36.1 |
| | +L | 52.6 | 53.2 | 59.5 | 46.1 | 26.1 | 43.6 | 55.1 | 38.6 |
| | +K | 51.0 | 54.7 | 60.2 | 63.9 | 28.4 | 44.6 | 57.5 | 36.5 |
| | +K360 | 50.0 | 55.0 | 59.8 | 65.0 | 42.7 | 45.2 | 54.5 | 45.2 |
| | +W | 54.8 | 56.4 | 60.5 | 66.8 | 43.4 | 62.7 | 57.4 | - |

within the image, altering the contents of the image patch $I(u-s, v-s, u+s, v+s)$. Meanwhile, shifts in the ego frame influence the value of query $\mathbf{p}_0$ and object location $(x, y, z)$[4]. Observing identical 3D objects with different sensor configurations alters the distributions of both 2D features and 3D positions, yielding an inconsistent mapping function $\mathcal{H}$. Consequently, detectors make incorrect predictions during cross-dataset testing and learn from conflicting data samples in multi-dataset training. In summary, the sensor deviation between datasets is three-fold:

- **Intrinsic.** Variations in camera intrinsic parameters, particularly the focal length, cause objects of identical size and location to be rendered differently in images across datasets.
- **Extrinsic.** As indicated in Eq. (1), extrinsic parameters or camera poses, especially $t_x$, also impact the apparent size of the object in images.
- **Ego coordinate system.** Fluctuations in ego centers affect data distribution. Notable differences in ego height impair the reliability of query prior knowledge in cross-dataset testing.

These discrepancies are illustrated in Fig. 1, where similar 3D information corresponds to highly distinct 2D image information with changes in the sensor suite.

## 4 Sensor Alignment Approaches

We introduce three efficient strategies to tackle the challenges: Intrinsic Synchronization, Extrinsic Aware Feature Sampling and Ego Frame Alignment. We observe that implementing the last two without Intrinsic Synchronization leads to sub-optimal outcomes[5]. Our approaches collectively create a sensor-invariant 3D-2D mapping relationship, enhancing model consistency across diverse datasets.

### 4.1 Intrinsic Synchronization (K-sync)

Among the factors impacting model performance, camera intrinsic parameters prove the most straightforward yet crucial to synchronize. Inspired by the intrinsic-decoupled technique prevalent in depth estimation [31, 32], we resize the images to a fixed focal length, $f_0$, using bi-linear interpolation.

---

[4]Also known as the ground-truth labels.

[5]See section 5.4 in the Appendix for ablation studies on sensor alignment approaches.

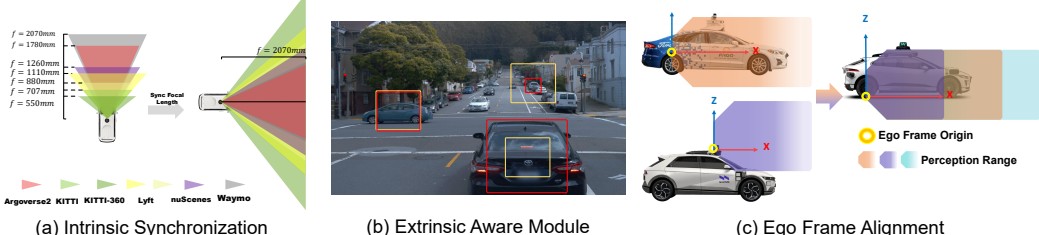

|  (a) Intrinsic Synchronization  |  (b) Extrinsic Aware Module  |  (c) Ego Frame Alignment  |

Figure 3: **(a)**: Resizing both input images and their focal length to achieve a unified focal length. **(b)**: Altering the fixed sampling region to vary in size, dependent on the distance between the camera center and query points(yellow to red). **(c)**: Aligning varied ego frames by adjusting the ego origin in accordance with the actual height and dataset distribution.

As depicted in Fig. 3(a), we align all focal lengths with that of Waymo, which has the largest focal length.

As shown in Table 2, simple resizing makes huge improvement. Compared to testing naively, over 24 mAP gains is achieved when transferring from nuScenes to Argoverse2 and KITTI. From KITTI-360 to KITTI, DETR3D attains a mAP of 34.6%, a result on par with in-domain evaluation. This enhancement aligns with the minimal domain gap between these two datasets, except for their focal length difference (707mm vs. 552mm). Table 3 displays the result of models trained on multiple datasets, where both in-domain and cross-domain performances exhibit substantial uplifts. The adverse effect previously associated with KITTI-360 is mitigated, signaling a resolution to the issue of conflicting data samples. Consequently, models are now efficiently utilizing the increased data volume to enhance performance.

### 4.2  Extrinsic Aware Feature Sampling (E-aware)

We introduce the Extrinsic Aware Feature Sampling (E-aware) to counteract the challenges posed by variations in camera extrinsics, specifically the frontal translation $t_x$. Our focus is on the impact of $t_x$ on the apparent object size $s_{pixel}$ and the related content within a fixed image receptive field $2s \times 2s$, as explained in Eq. (1) and Eq. (2).

Given the assumption that 3D query $\mathbf{p}_0$ as the object's 3D center, we modify the receptive field of $\mathbf{p}_0$ to be proportional to $\frac{c}{x_0 - t_x}$, ensuring that the sampled image content remains consistent across varying $t_x$. This modification is implemented by sampling more points followed by average pooling, analogous to the ROI Align process [38].

To validate the effectiveness of E-aware, we simulate changes in camera position through random translations within a range of $[-2m, 2m]$. Our method exhibits enhanced robustness to these positional fluctuations, as indicated in Table 4. The influence of $t_y$ is found to be minimal, while adjustments for $t_z$, are incorporated in the subsequent approach. Evaluations of E-aware under cross-dataset testing and multi-dataset training are also combined with the next approach.

### 4.3  Ego Frame Alignment (Ego-sync)

Variations in the definition of the ego frame across datasets, particularly the height of the ego centers, lead to inconsistencies in data distribution. Our initial strategy involves transforming the ego x-y plane to the ground, which ensures consistent physical interpretations of the z-coordinates across datasets. However, this straightforward approach yields no significant improvements. We turn to a nuanced strategy, employing DETR3D trained on the Waymo dataset as a reference. The ego center's x-z coordinates for each dataset are adjusted to align with this reference. A grid search, conducted at a resolution of $0.5m$, identifies the optimal alignment settings that maximize performance, as visualized in Fig. 4. We find that not only the height, but also the frontal position of the ego center play a pivotal role, since it influence the distribution of object's x-coordinates in each dataset.

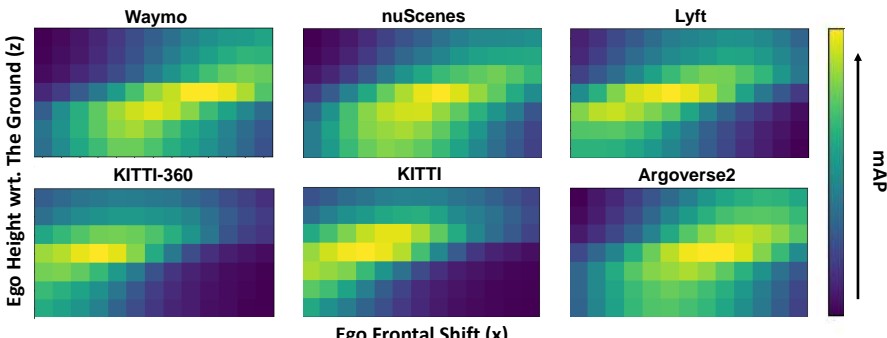

Figure 4: Exploring optimal performance through grid-search modifications of the x- and z-coordinates of ego centers, with the integration of K-sync and E-aware. The results indicate that both the height and frontal position of the ego center significantly impact performance metrics.

Table 4: Results of random jittering experiments on nuScenes, with the x-axis aligned to the car's direction of movement and the y-axis perpendicular to it.

| methods | None | on y | on x | on x,y |
|---|---|---|---|---|
| Direct | 36.3 | 34.7 | 27.9 | 26.9 |
| w/ E-aware | 36.9 | 35.5 | 35.4 | 34.9 |

The comprehensive alignment of ego frames, depicted in Fig. 3, yields enhanced performance metrics in Table 2 and Table 3. For instance, when training on six diverse datasets, DETR3D exhibits an mAP enhancement of up to 27.3% on KITTI compared to the baseline direct merging approach.

## 5    Conclusion

In this paper, we meticulously examined the obstacles hindering image-based detectors from delivering optimal performance and adaptability across various autonomous driving datasets. We pinpointed the root of the issue to the inconsistent 3D-2D mapping relationships, primarily caused by disparate sensor configurations encompassing camera intrinsic, extrinsic, and ego coordinate systems. We demonstrate that simple sensor alignment techniques can significantly alleviate this performance degradation. Our approach yielded an average enhancement of 29.5 mAP in cross-dataset testing from nuScenes to other datasets, capitalizing on nuScenes' diverse data distribution. We also achieve 21, 17.2, 6.3, 18.4, and 24.2 mAP boosts when AV2, Lyft, KITTI, KITTI-360 and Waymo serve as the source domain. Unlike many existing studies that only focus on vehicles, our evaluation metric also takes into account pedestrians and bicycles, offering a more comprehensive assessment.

In multi-dataset training, we fully exploit the potential of data volume, with 18.5, 8.4, 23.2, 42.3, 17.3, and 3.9 mAP gaining by combining 6 datasets instead of training on them separately. Compared to direct merging, we achieve an average performance boost of more than 10 mAP on all datasets. We believe that our insights will stimulate further research in multi-dataset training and domain adaptation for vision-centric 3D object detection and localization. We emphasize the importance of applying data corrections before incorporating additional datasets or developing new computer vision algorithms to bridge the remaining domain gaps.

## 6    Limitations

While our study provides valuable insights into addressing sensor misalignment in image-based detectors for autonomous driving, there are two limitations to consider. First, due to the lack of datasets with highly diverse camera poses, we were unable to explore the impact of camera rotation on the detection performance. Second, We were unable to scale our method to a larger-scale in-the-wild

dataset due to annotation scarcity. A potential solution could be employing semi-supervised 3D detectors as baselines, and we leave it to future work.

**Acknowledgments**

This work is supported by the National Key R&D Program of China (2022ZD0161700).

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

# Supplementary materials of Cross-dataset Sensor Alignment: Making Visual 3D Object Detector Generalize

**Anonymous Author(s)**
Affiliation
Address
`email`

1 Section 1: brief introduction to the six datasets.

2 Section 2: data format conversion and dataset merging settings.

3 Section 3: details of training settings.

4 Section 4: analysis and results of BEVDet.

5 Section 5: additional results from DETR3D.

## 1 Datasets

7 **Argoverse2.** The Argoverse2 dataset [1] is collected across six cities in the U.S., including Pittsburgh,
8 Detroit, Austin, Palo Alto, Miami, and Washington D.C. It encompasses data captured in various
9 weather conditions and at different times of the day. The dataset includes images from two grayscale
10 stereo cameras and seven cameras that provide 360-degree coverage. It offers 3D annotations at
11 a frame rate of 10Hz. To align with the frame rate in the nuScenes dataset, we sub-sample the
12 Argoverse2 dataset, resulting in 21,982 frames for training and 4,705 for validation, with a frame rate
13 of 2Hz.

14 **KITTI.** The KITTI [2] object detection benchmark consists of 7,481 frames for training. These
15 scenes were captured in clear weather and during daytime around Karlsruhe, Germany. The dataset
16 provides images from two RGB cameras and two grayscale cameras, forming two stereo pairs. For
17 our study, we solely utilize the left RGB camera. Following [3], we separate the data into 3,712
18 training frames and 3,769 validation frames.

19 **KITTI-360.** The KITTI-360 dataset [4] is significantly larger than KITTI, comprising 61,569 valid
20 frames with 3D annotations at a frame rate of 10Hz. The labelled data is obtained from nine video
21 clips. To create a training and validation split, we utilize the first 80% of each video clip for training
22 and the remaining 20% for validation. This results in a training set containing 49,253 frames and a
23 validation set containing 12,316. Unlike KITTI, the KITTI-360 dataset provides RGB images from
24 two frontal perspective cameras and two side fish-eye cameras. Similar to the KITTI settings, we
25 exclusively use the images from the left frontal camera in our study.

26 **nuScenes.** The nuScenes dataset [5] contains 28130 training and 6019 validation keyframes. The
27 scenes are collected around Boston, USA and Singapore in multiple weathers and during different
28 time frames. For each frame, the dataset provides six images that collectively cover a 360-degree
29 view.

30 **Lyft.** The Lyft Level 5 dataset [6] consists of 22,680 annotated frames captured around Palo Alto,
31 USA, during clear weather conditions and daytime. Each frame within the dataset includes images
32 from six surrounding view cameras as well as a long-focal-length frontal camera. It is essential to
33 mention that this dataset is collected using 20 independent vehicles, and the surrounding view images

have two different resolutions. Following the approach outlined in MMDetection3D [7], we partition the dataset into 18,900 frames for training and 3,780 frames for validation.

**Waymo.** The Waymo [8] dataset is collected across Phoenix, Mountain View, and San Francisco, encompassing various weather conditions and different times of the day. It includes images from five cameras and offers 3D annotations at a frame rate of 10Hz. The training set consists of 158,081 frames, while the validation set contains 39,990 frames. To align with a 2Hz frame rate, we sub-sample the dataset, resulting in 31,616 frames for training and 7,998 frames for validation.

## 2 Converting Datasets into a Unified Format

This section provides a detailed explanation of how we convert Argoverse2, KITTI, KITTI-360, Lyft, nuScenes, and Waymo datasets into a unified format. We specifically focus on the issues related to coordinate systems and 3D annotations that arise when merging these datasets. We convert data under MMDetection3D v1.1.0.

### 2.1 Coordinate Systems

Regarding sensor configuration, the datasets differ in terms of three types of coordinate systems: ego frame, LiDAR frame, and camera frame. The definition of camera is clear, so we primarily focus on the former two. Each dataset typically includes at least one LiDAR mounted on the vehicle's roof. The origin of the LiDAR frame is commonly located at the center of the top LiDAR if there is no specification.

The ego frame is more confusing as the origin is defined differently across the datasets. In Argoverse2, nuScenes and Waymo, the ego origin is located at the center of the car's rear axle. In Argoverse2, it is approximately 33cm above the ground, while in the latter two datasets, it is projected onto the ground plane. Lyft does not explicitly specify the location; however, based on the statistical analysis of 3D annotations, it is also considered on the ground. These four datasets have corrected their axes, ensuring the z-axis consistently points upwards from the road surface. On the other hand, for KITTI and KITTI-360, the Inertial Measurement Unit (IMU) defines the ego frame. Across all the datasets, the x-axis aligns with the car's longitudinal direction, while the y-axis points to the left.

Regarding LiDAR point clouds and 3D annotations, Argoverse2 and Waymo define them in the ego frame, while KITTI, KITTI-360, Lyft, and nuScenes define them in the LiDAR frame. Consequently, during training, we consider the LiDAR centers of the latter datasets as the 'ego centers'.

In terms of ego frame alignment, for Argoverse2, KITTI, and KITTI-360, we simply lower their ego centers by 0.33m, 1.73m, and 1.73m, respectively, to align them with the road surface. For Lyft and nuScenes, we transform the entire coordinate system to their original ego frames, which are also pressed against the road.

### 2.2 Object Filtering

To ensure consistency and data quality, we discard object annotations that fall outside the camera view. This is accomplished by projecting the eight corners of each object's 3D bounding box onto the image plane. The object annotation is removed if all eight corners are outside the image boundary. Additionally, we filter annotations based on a specific range in the x, y, and z coordinates, namely $[-51.2, 51.2] \times [-51.2, 51.2] \times [-5.0, 4.0]$. As every dataset includes LiDAR data, we also discard annotations with no LiDAR points within the 3D bounding box since they may be occluded.

### 2.3 Merging Categories

To unify the category labels across datasets, we merge the categories within each dataset into three classes: vehicle, pedestrian, and bicycle. This taxonomy closely resembles Waymo's classification but with a little difference in the bicycle category. Waymo excludes bicycles without a rider, whereas we

Table 1: The original categories in each dataset vs. merged categories

| Dataset | Vehicle | Pedestrian | Bicycle |
|---------|---------|------------|---------|
| Argoverse2 | REGULAR VEHICLE, LARGE VEHICLE, BUS, BOX TRUCK, TRUCK, MOTORCYCLE, VEHICULAR TRAILER, TRUCK CAB, SCHOOL BUS | PEDESTRIAN, WHEELED RIDER, OFFICIAL SIGNALER | BYCYCLE, BYCYCLIST |
| KITTI | Car, Van, Trunk, Tram | Pedestrian, Person Sitting | Cyclist |
| KITTI-360 | bus, car, caravan, motorcycle, trailer, train, truck, unknown vehicle | person | bicycle, rider |
| Lyft | car, truck, bus, emergency vehicle, other vehicle, motorcycle | pedestrian | bicycle |
| nuScenes | car, truck, construction vehicle, bus, trailer, motorcycle | pedestrian | bicycle |
| Waymo | Car | Pedestrian | Cyclist |

Table 2: Percentage of each class in each dataset.

| Dataset | Vehicle | Pedestrian | Cyclist |
|---------|---------|------------|---------|
| N | 70.3% | 26.6% | 3.1% |
| A | 71.9% | 26.9% | 1.2% |
| L | 93.4% | 3.7% | 2.9% |
| K | 84.8% | 11.4% | 3.8% |
| K360 | 89.9% | 4.9% | 5.1% |
| W | 64.3% | 34.7% | 1.0% |

include such objects when relabeling the other datasets. Table 1 shows the mapping of all categories to the three classes. Any types not listed in the table are discarded during the merging process. Table 2 also shows the percentage of each class in each dataset.

## 3  Training Details

For all detectors, the image backbone is a Resnet-50 [9] pretrained on ImageNet [10].

**DETR3D.** We use images with original resolution and the original training policy [11].

**BEVDet.** The input images are at 1/2 width and height. We use adamW [12] with weight decay $1 \times 10^{-7}$ as optimizer, and train it for 24 epochs with batch size 64 and initial learning rate $2 \times 10^{-4}$, which will be decreased 10 times on 20th and 24th epoch. We also set the depth bins to be $[1, 140]$.

During training, we deploy data augmentations in BEV space and image input for BEVDet. We follow the original settings in [13], except that we do random scaling with a range of $[0.8, 1.2]$. After scaling, we crop the bottom part of the images from every dataset and pad them to a unified resolution: $960 \times 448$.

## 4  Analysis and Results of BEVDet

We have verified our sensor alignment strategy on BEVDet, which leverages depth estimation to shoot 2D image features to 3D space. The results are shown in Table 3 and Table 4. Since BEVDet represents another type of BEV detector against DETR3D (LSS-based vs. Query-based), there are some differences in the alignment approaches.

### 4.1  Sensor Alignment Approaches

**Intrinsic Synchronization (K-sync).** Because BEVDet requires inputs of fixed resolution, image resizing is not very compatible since it introduces dynamic resolution. Instead, we sync focal length by applying intrinsic-decoupled depth estimation as [14, 15, 16] do. It is confusing that the performance ("K-sync" in Table 3 and Table 4) does not improve much. We visualize the prediction in Fig. 1. Surprisingly, this module predicts depth correctly. The reason for failure is the wrong heights.

Table 3: 3D-mAP of BEVDet [13] trained on single dataset. "Direct" means direct transfer, "K-sync" means Intrinsic Synchronization ("K" stands for the intrinsic matrix **K**), and "Ego-sync" means Ego Frame Alignment.

| Setting | src\dst | N | A | L | K | K360 | W | avg.T |
|---|---|---|---|---|---|---|---|---|
| Direct | N | 29.5 | 0.0 | 0.3 | 0.0 | 0.0 | 0.0 | 0.1 |
| | A | 0.0 | 34.3 | 0.0 | 0.0 | 0.0 | 9.0 | 1.8 |
| | L | 0.0 | 0.0 | 31.8 | 0.1 | 0.2 | 0.0 | 0.1 |
| | K | 0.0 | 0.0 | 0.1 | 9.2 | 0.0 | 0.0 | 0.0 |
| | K360 | 0.0 | 0.0 | 0.0 | 0.2 | 19.5 | 0.0 | 0.0 |
| | W | 0.0 | 0.4 | 0.0 | 0.0 | 0.0 | 45.1 | 0.1 |
| K-sync | N | 30.7 | 0.2 | 3.9 | 5.0 | 1.3 | 0.5 | 2.2 |
| | A | 2.4 | 37.9 | 0.0 | 0.0 | 0.0 | 4.2 | 1.3 |
| | L | 0.2 | 0.0 | 32.0 | 0.3 | 0.6 | 0.0 | 0.2 |
| | K | 0.2 | 0.0 | 0.1 | 10.7 | 1.5 | 0.0 | 0.4 |
| | K360 | 0.0 | 0.1 | 2.7 | 11.0 | 18.3 | 0.0 | 2.8 |
| | W | 0.3 | 17.8 | 0.0 | 0.0 | 0.0 | 47.6 | 3.6 |
| K-sync + Ego-sync | N | 31.9 | 9.8 | 2.8 | 6.3 | 3.7 | 0.6 | 4.6 |
| | A | 7.0 | 37.8 | 4.6 | 2.9 | 1.2 | 3.1 | 3.8 |
| | L | 0.5 | 0.0 | 31.6 | 2.6 | 3.5 | 0.0 | 1.3 |
| | K | 0.2 | 0.3 | 1.3 | 10.0 | 1.6 | 0.2 | 0.7 |
| | K360 | 0.0 | 0.0 | 3.4 | 12.4 | 21.3 | 0.0 | 3.2 |
| | W | 11.6 | 26.3 | 8.8 | 10.8 | 3.9 | 47.6 | 12.3 |

Table 4: 3D-mAP of BEVDet trained on **multiple** datasets. "Direct" means direct merge and transfer.

| Setting | src\dst | N | A | L | K | K360 | W | avg.S | avg.T |
|---|---|---|---|---|---|---|---|---|---|
| Direct | N | 29.5 | 0.0 | 0.3 | 0.0 | 0.0 | 0.0 | 29.5 | 0.1 |
| | +A | 33.7 | 38.8 | 0.0 | 0.1 | 0.0 | 8.7 | 36.2 | 2.2 |
| | +L | 36.2 | 40.0 | 38.2 | 0.1 | 0.1 | 4.5 | 38.1 | 1.6 |
| | +K | 32.1 | 40.3 | 37.7 | 29.9 | 0.2 | 5.7 | 35.0 | 3.0 |
| | +K360 | 33.4 | 40.0 | 39.0 | 32.5 | 23.4 | 8.7 | 33.7 | 8.7 |
| | +W | 30.8 | 37.7 | 38.4 | 34.6 | 22.3 | 45.2 | 34.8 | - |
| K-sync | N | 30.7 | 0.2 | 3.9 | 5.0 | 1.3 | 0.5 | 30.7 | 2.2 |
| | +A | 34.8 | 41.2 | 1.4 | 1.2 | 0.4 | 4.3 | 38.0 | 1.8 |
| | +L | 37.1 | 40.5 | 38.7 | 4.6 | 3.6 | 16.8 | 38.8 | 8.3 |
| | +K | 38.7 | 41.1 | 38.5 | 32.9 | 7.1 | 12.2 | 37.8 | 9.6 |
| | +K360 | 35.3 | 42.5 | 39.4 | 36.9 | 23.2 | 6.4 | 35.5 | 6.4 |
| | +W | 36.7 | 46.8 | 39.2 | 38.3 | 25.2 | 52.3 | 39.8 | - |
| K-sync + Ego-sync | N | 31.9 | 9.8 | 2.8 | 6.3 | 3.7 | 0.6 | 31.9 | 4.6 |
| | +A | 36.8 | 41.2 | 7.2 | 11.8 | 2.5 | 3.2 | 39.0 | 6.2 |
| | +L | 38.7 | 44.0 | 36.4 | 16.4 | 10.3 | 2.6 | 39.7 | 9.8 |
| | +K | 37.5 | 41.5 | 38.8 | 34.2 | 11.0 | 9.4 | 38.0 | 10.2 |
| | +K360 | 39.4 | 43.9 | 38.4 | 40.2 | 27.3 | 2.7 | 37.8 | 2.7 |
| | +W | 40.8 | 46.1 | 39.7 | 42.7 | 28.1 | 51.4 | 41.5 | - |

When we use a BEV metric that ignores heights or executes Ego Frame Alignment, the performance improves immediately. See § 4.2 and § 4.3 for more details of BEV metric and intrinsic-decoupled depth estimation.

**Extrinsic Aware Feature Sampling (E-aware).** This module is no longer applicable since the depth estimation module does not predict depths from the ego center. On the contrary, it predicts from the camera optical center and naturally bypasses the impact of extrinsic. However, even if the depth of image features are correctly inferred, we hypothesis that model still struggles in estimating the dimensions of objects, which may be addressed by introducing extrinsic embedding.

**Ego Frame Alignment (Ego-sync).** We use the same Ego-sync settings as in the main paper. In Table 3, transferring from Waymo to other datasets has an improvement of 12.2 in 3D-mAP. In

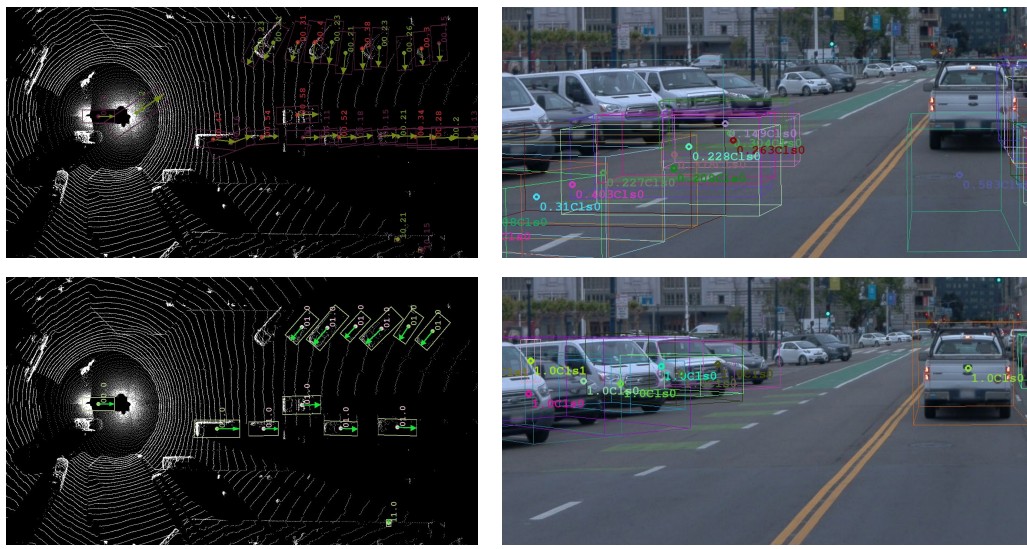

Figure 1: Visualization of objects in point clouds and images. The top is prediction made by BEVDet and the bottom shows groundtruths.

Table 4, BEVDet achieves a performance boost of 6.7 mAP when training on all datasets. The similar mAP gains proves the importance of ego frame definition once again.

## 4.2 3D Metric vs. BEV Metric

The poor cross-dataset improvement after applying K-sync seems inconsistent with the one we gained in DETR3D. However, after careful examination, we find that although K-sync corrects the depth prediction, it cannot correct object heights.

The visualization of detection results proves this point. As shown in Fig. 1, in bird-eye view, the predicted objects have become very close to the ground truth after we apply K-sync. However, a large but consistent offset appears on the image plane: the objects are predicted lower than they should be. Although the depths are estimated correctly, the heights are wrong, causing the 3D mAP to be nearly zero.

We show the power of K-sync by changing our 3D metric into a BEV metric. In Table 5, we switch the metric by setting the center heights to zero for both prediction and ground truth objects. A significant improvement emerges after we apply both BEV metric and K-sync, up to 10 points.

The reason is simple: K-sync only corrects depths. In DETR3D, objects are inferred from 3D query points that contain height information, while BEVDet collapses 3D grid into BEV pillars, losing the height information. LSS-based [17] methods often assume all objects to be on flat ground. Under this setting, it ignores height information and can not deal with changes in altitude, such as irregular terrains and ego frame changes. This could be solved by setting 3D voxel grid instead of BEV pillars, or post-process the heights according to the terrain.

## 4.3 Intrinsic Decoupled Module

The dense depth estimation module in BEVDet requires a fixed input resolution throughout training and testing, which prevents BEVDet from changing the input resolution as flexibly as DETR3D can. As an alternative, We scale the predicted depth according to the focal length. We call it Intrinsic Decoupled Module.

Table 5: **BEV-mAP** of BEVDet [13] trained on single dataset with Intrinsic Synchronization. We show avg.T improvement compared to 3D-mAP

| Setting | src/dst | N | A | L | K | K360 | W | $\Delta$ avg.T |
|---|---|---|---|---|---|---|---|---|
| K-sync | N | 33.6 | 22.5 | 8.7 | 12.4 | 4.4 | 16.8 | +10.8 |
| | A | 13.6 | 39.7 | 7.2 | 8.0 | 3.4 | 21.6 | +9.5 |
| | L | 3.0 | 0.9 | 34.3 | 3.2 | 5.0 | 1.7 | +2.6 |
| | K | 0.9 | 2.5 | 3.2 | 13.4 | 2.7 | 1.1 | +1.7 |
| | K360 | 0.2 | 0.5 | 9.5 | 13.4 | 22.4 | 0.3 | +2.0 |
| | W | 16.1 | 29.1 | 10.2 | 10.6 | 3.7 | 50.3 | +10.3 |

Assuming that we have a depth estimation network, which is trained on a dataset with fixed focal length (e.g. nuScenes with f=1266), it will predict objects' depth according to their pixel size. Given $s_{pixel}$ as the pixel size of a certain object and $d$ as the metric depth (in meters), the network learns a mapping:

$$\mathcal{M} : s_{pixel} \longrightarrow d. \tag{1}$$

Intuitively, if an object looks small, it predicts a large depth, and vice versa, and if we resize the image to be smaller, all the objects look smaller, so the predicted depths increase. However, the 3D locations of objects do not change with image resizing, so predictions are eventually wrong.

We want the model learns and predicts a mapping that is invariant to the focal length changes, so we set a scale-invariant depth $d^*$:

$$d^* = \frac{f}{f_0} \times d, \tag{2}$$

where $f$ is the input focal length, and $f_0$ is a constant. It can be understood as the "reference focal length", *i.e.*, let the depth network "feel" as if it were working on a single camera, although it receives images of various focal lengths from different datasets.

In practice, we force the model to learn:

$$\mathcal{M}^* : s_{pixel} \longrightarrow d^*, \tag{3}$$

and we recover the metric depth by:

$$d = \frac{f_0}{f} \times d^*, \tag{4}$$

for shooting image features to BEV grid later.

## 5 Additional Results from DETR3D

In this section, we first provide additional results on monocular 3D detection using DETR3D.

### 5.1 Ablation Studies by Cropping the Input Images

To investigate whether the model relies on other visual cues for object detection, we conduct an experiment where we crop the input images at different positions during testing. In Table 6, we find no performance drop in DETR3D, whereas BEVDet shows a substantial decrease. This indicates that DETR3D does not rely on objects' position in the images. On the other hand, BEVDet, which incorporates a depth network in its architecture, relies more on this kind of pictorial cues, as suggested in prior work [18].

Table 6: Results of DETR3D and BEVDet which are trained on Waymo, using images cropped at different positions during testing.

| Cropped height | [192,992] | [288,1088] | [384,1184] | [480,1280] | Origin |
|---|---|---|---|---|---|
| DETR3D | 56.9 | 59.3 | 59.3 | 59.7 | 57.7 |
| BEVDet | 20.8 | 30.5 | 36.4 | 31.5 | 39.0 |

Table 7: Ablation study of synchronizing focal length to different values. We add "*" to indicate the original focal lengths.

| Train | focal length | N | A | L | K | K360 | W |
|---|---|---|---|---|---|---|---|
| N | not synced | 36.3 | 0.8 | 1.8 | 0.0 | 0.0 | 1.1 |
| | 1260* | 35.7 | 21.9 | 13.8 | 27.1 | 16.9 | 19.2 |
| | 2070 | 40.8 | 25.5 | 18.6 | 29.7 | 18.0 | 23.4 |
| | 3100 | 41.7 | 26.2 | 18.7 | 28.5 | 17.7 | 25.8 |
| | 4140 | 43.3 | 26.4 | 19.4 | 31.8 | 17.5 | 26.3 |
| A | not synced | 0.2 | 48.0 | 0.1 | 0.0 | 0.0 | 17.4 |
| | 1780* | 11.8 | 46.8 | 7.2 | 6.4 | 5.4 | 38.4 |
| | 2070 | 13.2 | 51.4 | 7.5 | 6.6 | 4.6 | 38.8 |
| | 3100 | 12.1 | 51.1 | 8.9 | 7.5 | 5.1 | 40.7 |
| | 4140 | 11.5 | 53.9 | 9.2 | 6.1 | 4.0 | 40.9 |
| L | not synced | 0.5 | 0.1 | 37.3 | 0.4 | 0.0 | 0.1 |
| | 2070 | 1.0 | 1.3 | 44.0 | 8.1 | 5.7 | 1.5 |
| | 3100 | 1.1 | 1.3 | 41.0 | 8.2 | 4.8 | 1.4 |
| | 4140 | 1.0 | 1.6 | 42.9 | 10.5 | 3.2 | 1.6 |
| K360 | not synced | 0.1 | 0.2 | 0.0 | 3.2 | 26.1 | 0.1 |
| | 550* | 8.2 | 4.5 | 3.3 | 18.4 | 25.9 | 5.5 |
| | 2070 | 14.6 | 14.7 | 7.3 | 34.6 | 34.7 | 8.2 |
| | 3100 | 14.0 | 15.4 | 9.4 | 33.6 | 35.9 | 7.5 |
| W | not synced | 0.1 | 8.9 | 0.0 | 0.0 | 0.0 | 58.8 |
| | 2070* | 14.5 | 37.8 | 14.3 | 9.4 | 5.6 | 57.7 |
| | 3100 | 14.6 | 38.1 | 13.6 | 7.7 | 3.2 | 62.6 |
| | 5170 | 10.1 | 38.7 | 10.7 | 8.8 | 2.0 | 62.1 |

## 5.2 Synchronize Focal Length to Different Values

This subsection shows that our Intrinsic Synchronization strategy works with different focal lengths. We perform an ablation study by increasing the synchronized focal length value in both training and testing. In Table 7, we observe that enlarging the images does improve mAP; however, the extent of improvement diminishes as the input image size increases. We argue that this phenomenon can be attributed to the fact that smaller objects become easier to detect in larger images.

## 5.3 Ablation Studies on Sensor Alignment Approaches

We evaluate various combinations of modules across all datasets and present the results in Table 8. We observe that each component contributes to the overall performance; however, only after aligning the intrinsic parameters does the extrinsic and ego coordinate system start to impact the detection performance. While the Extrinsic Aware Feature Sampling (E-aware) may cause a drop in performance, we argue that this module provides extrinsic robustness in real-world scenarios.

## 5.4 Multi-dataset Training Beginning with Waymo

We begin with Waymo as the first dataset and gradually add datasets into the training set. Table 9 shares the same mAP trend with Table 3 in the main paper, which means our observation is invariant to the addition order. Here, KITTI-360 drags down the performance again. This decline can be

Table 8: Ablation study on the effectiveness of each module in sensor alignment approaches. All models are trained on Waymo. "Ego" stands for Ego Frame Alignment. "Avg.T" stands for the average cross-dataset performance.

| K-sync | E-aware | Ego | N | A | L | K | K360 | W | avg.T |
|---|---|---|---|---|---|---|---|---|---|
| | | | 0.1 | 8.9 | 0.0 | 0.0 | 0.0 | 58.8 | 1.8 |
| | | ✓ | 0.0 | 8.4 | 0.0 | 0.0 | 0.0 | 58.3 | 1.7 |
| | ✓ | | 0.0 | 5.0 | 0.0 | 0.0 | 0.0 | 59.4 | 1.0 |
| | ✓ | ✓ | 0.0 | 3.5 | 0.0 | 0.0 | 0.0 | 59.4 | 0.7 |
| ✓ | | | 14.5 | 37.8 | 14.3 | 9.4 | 5.6 | 57.7 | 16.3 |
| ✓ | | ✓ | 24.7 | 39.4 | 33.0 | 21.2 | 13.6 | 57.7 | 26.4 |
| ✓ | ✓ | | 14.1 | 37.7 | 17.0 | 9.3 | 5.8 | 57.6 | 16.8 |
| ✓ | ✓ | ✓ | 25.4 | 38.2 | 33.6 | 21.2 | 11.7 | 57.6 | 26.0 |

Table 9: 3D-mAP of DETR3D trained on multiple dataset, beginning with Waymo.

| Setting | src/dst | W | N | A | L | K | K360 | avg.S | avg.T |
|---|---|---|---|---|---|---|---|---|---|
| Direct | W | 58.8 | 0.1 | 8.9 | 0.0 | 0.0 | 0.0 | 58.8 | 1.8 |
| | +N | 60.4 | 38.4 | 11.2 | 0.2 | 0.0 | 0.0 | 49.4 | 2.8 |
| | +A | 63.0 | 42.7 | 54.8 | 0.1 | 0.0 | 0.0 | 53.5 | 0.0 |
| | +L | 60.2 | 44.4 | 53.1 | 47.3 | 0.0 | 0.0 | 51.2 | 0.0 |
| | +K | 63.1 | 45.5 | 53.4 | 49.2 | 44.3 | 1.9 | 51.1 | 1.9 |
| | +K360 | 61.9 | 46.2 | 53.7 | 49.4 | 39.5 | 29.7 | 46.7 | 0.0 |

attributed to a significant amount of discordant data, as illustrated in Fig. 2, which provides statistics on the data volume across different datasets.

## 5.5 Per-class and Per-location Evaluation Results

Table 10 and Table 11 are the extended version of Table 2 and Table 3 in the main paper, showing per-class mAP on each dataset. Furthermore, Table 12 shows the evaluation result of the best model in Table 3 on each city in each dataset.

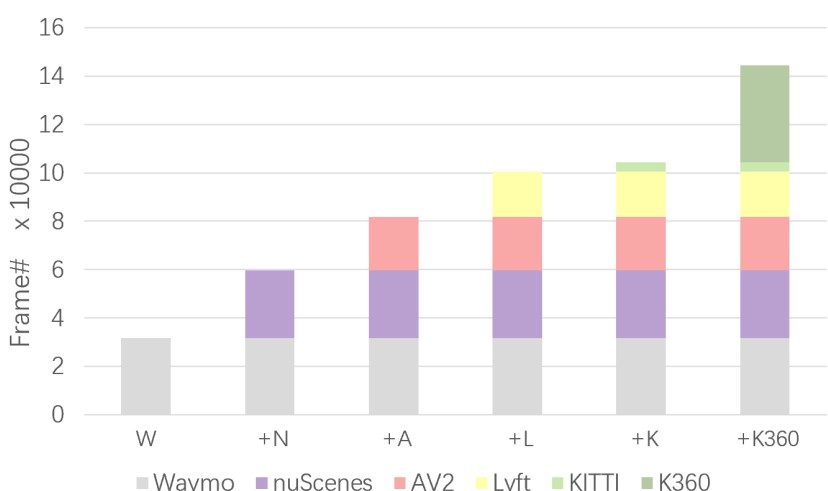

Figure 2: Data volume of each dataset under monocular detection setting

Table 10: 3D-mAP of DETR3D trained on single dataset(full version). The performance is reported in the format of all(vehicles/ pedestrians/ bicycles).

| Setting | src/dst | N | A | L | K | K360 | W |
|---|---|---|---|---|---|---|---|
| Direct | N | 36.3 (56.7/36.7/15.7) | 0.8 (0.9/1.4/0.3) | 1.8 (1.6/1.1/2.7) | 0.0 (0.0/0.0/0.0) | 0.0 (0.0/0.0/0.0) | 1.1 (0.6/1.2/1.4) |
| | A | 0.2 (0.1/0.5/0.1) | 48.0 (73.8/38.7/31.7) | 0.1 (0.0/0.1/0.1) | 0.0 (0.0/0.1/0.0) | 0.0 (0.0/0.1/0.0) | 17.4 (19.7/14.1/18.3) |
| | L | 0.5 (0.9/0.7/0.0) | 0.1 (0.1/0.1/0.0) | 37.3 (70.5/16.6/24.9) | 0.4 (0.5/0.7/0.1) | 0.0 (0.0/0.1/0.0) | 0.1 (0.0/0.3/0.0) |
| | K | 2.8 (4.8/3.0/0.5) | 1.2 (0.9/1.3/1.5) | 0.0 (0.1/0.1/0.0) | 24.5 (40.2/25.1/8.3) | 1.1 (0.9/2.1/0.4) | 0.7 (0.2/0.4/1.3) |
| | K360 | 0.1(0.1/0.2/0.0) | 0.2(0.0/0.2/0.4) | 0.0(0.0/0.0/0.0) | 3.2(0.9/6.7/2.2) | 26.1(60.2/4.5/13.7) | 0.1(0.0/0.1/0.2) |
| | W | 0.1 (0.0/0.1/0.0) | 8.9 (14.5/9.2/3.1) | 0.0 (0.0/0.0/0.0) | 0.0 (0.0/0.0/0.0) | 0.0 (0.0/0.0/0.0) | 58.8 (78.1/50.3/47.9) |
| K-sync | N | 40.8 (58.5/42.1/21.8) | 25.5 (45.6/25.1/5.7) | 18.6 (39.8/14.4/1.6) | 29.7 (41.1/24.2/23.7) | 18.0 (37.6/11.3/4.9) | 23.4 (37.4/24.8/7.9) |
| | A | 13.2 (20.4/13.7/5.5) | 51.4 (74.8/43.5/36.0) | 7.5 (16.7/3.9/1.9) | 6.6 (8.9/1.8/9.1) | 4.6 (12.2/0.8/0.8) | 38.8 (60.4/31.3/24.6) |
| | L | 1.0 (1.8/1.2/0.1) | 1.3 (3.3/0.4/0.1) | 44.0 (76.4/20.9/34.6) | 8.1 (17.3/4.7/2.3) | 5.7 (12.7/2.0/2.4) | 1.5 (3.2/1.0/0.2) |
| | K | 2.4 (2.0/4.9/0.4) | 1.2 (2.6/0.8/0.1) | 1.2 (3.0/0.5/0.1) | 31.0 (45.7/28.0/19.4) | 6.1 (13.3/2.7/2.5) | 0.5 (1.2/0.2/0.1) |
| | K360 | 14.6 (34.0/9.3/0.4) | 14.7 (36.4/2.7/4.9) | 7.3 (20.0/1.6/0.2) | 34.6 (59.9/25.1/18.9) | 34.7 (69.4/10.0/24.7) | 8.2 (22.7/0.7/1.1) |
| | W | 14.5(25.9/13.3/4.4) | 37.8(67.3/34.1/11.9) | 14.3(25.5/8.5/8.8) | 9.4(6.3/4.8/17.0) | 5.6(13.1/1.1/2.6) | 57.7(78.0/50.2/45.0) |
| K-sync, E-aware, and Ego-sync | N | 43.1(63.0/45.0/21.4) | 33.6(62.4/30.4/7.9) | 32.8(60.8/19.3/18.4) | 33.0(49.0/28.2/21.7) | 18.4(37.0/12.8/5.5) | 33.0(47.8/28.4/22.8) |
| | A | 24.4(42.0/23.0/8.3) | 48.1(75.3/41.7/27.3) | 34.1(61.5/22.4/18.3) | 18.1(22.0/17.3/15.0) | 8.7(16.1/3.1/6.8) | 37.4(58.4/28.8/24.9) |
| | L | 15.7(35.1/10.3/1.9) | 19.6(42.6/13.1/3.0) | 47.1(79.3/21.2/40.8) | 20.0(38.7/11.5/9.9) | 12.9(31.8/4.0/2.9) | 18.9(30.2/11.2/15.2) |
| | K | 7.1(12.6/7.9/0.8) | 8.7(19.2/3.8/2.9) | 10.2(20.7/4.2/5.6) | 29.1(49.5/28.1/9.6) | 9.3(20.2/4.9/2.7) | 2.4(3.6/2.5/1.1) |
| | K360 | 13.9(33.2/8.3/0.3) | 17.7(41.7/5.6/5.8) | 16.6(38.4/4.2/7.3) | 39.1(67.6/26.5/23.2) | 36.7(72.4/10.9/26.6) | 8.4(18.2/2.8/4.3) |
| | W | 25.4(45.0/26.3/5.0) | 38.2(68.3/34.6/11.7) | 33.6(63.3/16.8/20.7) | 21.2(13.5/25.3/24.7) | 11.7(25.6/6.2/3.3) | 57.6(77.9/50.9/44.2) |

Table 11: 3D-mAP of DETR3D trained on **multiple** dataset, beginning with nuScenes (full version). The performance is reported in the format of all(vehicles/ pedestrians/ bicycles)

| Setting | src/dst | N | A | L | K | K360 | W |
|---|---|---|---|---|---|---|---|
| Direct | N | 36.3 (56.7/36.7/15.7) | 0.8 (0.9/1.4/0.3) | 1.8 (1.6/1.1/2.7) | 0.0 (0.0/0.0/0.0) | 0.0 (0.0/0.0/0.0) | 1.1 (0.6/1.2/1.4) |
| | +A | 40.5 (60.4/38.9/22.1) | 49.2 (76.0/42.3/29.4) | 0.5 (0.7/0.5/0.3) | 0.0 (0.0/0.0/0.0) | 0.0 (0.0/0.0/0.0) | 5.2 (4.5/4.2/6.9) |
| | +L | 41.6 (61.0/40.7/23.1) | 50.5 (78.4/42.3/30.9) | 43.7 (74.5/26.3/30.4) | 0.0 (0.0/0.0/0.0) | 0.0 (0.0/0.0/0.0) | 3.8 (4.2/4.2/2.9) |
| | +K | 41.5 (62.7/39.6/22.1) | 49.7 (78.5/42.4/28.3) | 46.0 (75.8/28.1/34.2) | 41.4 (62.1/38.7/23.5) | 1.1 (1.2/2.1/1.0) | 3.6 (4.3/4.2/2.2) |
| | +K360 | 42.6 (64.2/40.8/22.8) | 54.3 (78.6/43.8/40.6) | 46.8 (76.7/26.5/37.1) | 36.3 (51.3/35.0/22.7) | 29.7 (60.6/8.6/20.0) | 3.3 (4.2/3.8/1.8) |
| | +W | 46.2 (66.6/42.7/29.2) | 53.7 (79.8/47.9/33.5) | 49.4 (76.7/33.0/38.4) | 39.5 (54.2/36.1/28.0) | 29.7 (60.6/9.4/19.2) | 61.9 (82.0/55.4/48.2) |
| K-sync | N | 40.8 (58.5/42.1/21.8) | 25.5 (45.6/25.1/5.7) | 18.6 (39.8/14.4/1.6) | 29.7 (41.1/24.2/23.7) | 18.0 (37.6/11.3/4.9) | 23.4 (37.4/24.8/7.9) |
| | +A | 45.5 (64.5/45.0/27.0) | 50.0 (77.9/44.0/28.1) | 25.1 (49.2/18.0/8.1) | 35.8 (48.1/26.2/33.1) | 21.3 (42.5/15.0/6.5) | 44.2 (67.9/35.8/28.7) |
| | +L | 46.8(64.3/47.1/28.9) | 53.2(79.5/46.6/33.6) | 55.1(82.6/36.4/46.2) | 37.8(50.7/30.6/31.9) | 23.1(44.6/16.9/7.8) | 45.3(69.2/37.0/29.6) |
| | +K | 47.4 (64.5/48.0/29.8) | 53.5 (79.3/45.6/35.6) | 53.6 (82.5/34.3/43.9) | 57.8 (77.7/48.7/46.9) | 21.8 (36.4/17.2/11.6) | 44.4 (69.4/37.2/26.7) |
| | +K360 | 50.2 (68.0/47.5/34.3) | 54.4 (80.7/48.3/34.3) | 54.0 (83.7/36.4/42.0) | 60.2 (81.9/47.2/51.4) | 39.6 (72.7/16.9/29.3) | 44.7 (71.6/38.2/24.2) |
| | +W | 51.8 (68.2/49.8/37.3) | 55.3 (82.1/48.8/34.9) | 56.6 (84.4/38.4/47.1) | 61.9 (83.0/51.5/51.1) | 40.7 (73.6/19.9/28.7) | 63.7 (83.2/55.0/52.8) |
| K-sync, E-aware, and Ego-sync | N | 43.1(63.0/45.0/21.4) | 33.6(62.4/30.4/7.9) | 32.8(60.8/19.3/18.4) | 33.0(49.0/28.2/21.7) | 18.4(37.0/12.8/5.5) | 33.0(47.8/28.4/22.8) |
| | +A | 52.1(68.1/50.8/37.4) | 52.7(77.9/47.3/32.9) | 38.4(70.4/23.5/21.2) | 42.2(54.9/33.6/37.9) | 23.2(43.0/16.5/10.3) | 40.7(64.1/35.6/22.5) |
| | +L | 52.6(68.9/50.2/38.6) | 53.2(79.1/47.4/33.2) | 59.5(85.6/45.5/47.4) | 46.1(61.6/37.8/38.9) | 26.1(47.6/19.7/10.9) | 43.6(67.1/35.6/28.0) |
| | +K | 51.0(67.9/51.8/33.3) | 54.7(79.8/47.7/36.5) | 60.2(85.6/44.5/50.5) | 63.9(83.2/58.0/50.6) | 28.4(48.8/22.9/13.5) | 44.6(67.1/35.4/31.4) |
| | +K360 | 50.0(70.5/50.4/29.3) | 55.0(81.4/47.4/36.2) | 59.8(86.9/43.9/48.4) | 65.0(85.4/54.5/55.1) | 42.7(75.5/20.3/32.3) | 45.2(68.6/36.2/30.9) |
| | +W | 54.8(72.7/52.5/39.1) | 56.4(82.3/49.0/38.0) | 60.5(87.4/45.7/48.4) | 66.8(85.2/58.1/57.2) | 43.4(76.3/22.3/31.5) | 62.7(83.4/56.9/47.9) |

## 5.6 Results from Surrounding-view Detection

We extend our data alignment strategies to surrounding view detection and verify their effectiveness. All input images are of 1/2 height and width. The perception range is still 51.2m, but includes the area behind the ego car.

**Ablation studies on sensor alignment approaches.** In Table 13, DETR3D is trained on Argoverse2, nuScenes and Waymo, and tested in six datasets.

**Data diversity vs. data volume** We test if the model benefits from data diversity more than data volume. Given that Waymo and nuScene are of similar data volume, we use different percentage of training data from the two and test the model on three datasets. As shown in Table 14, mixing the data achieves better performance.

Table 12: Evaluation results per location, using DETR3D with all sensor alignment approaches. The LET-3D-mAP is reported in the format of all(vehicles/pedestrians/bicycles).

| Dataset | Location | LET-3D-AP |
|---------|----------|-----------|
| Argoverse2 | ATX | 44.6 (69.7/64.3/0.0) |
| | DTW | 71.5 (79.3/60.3/75.1) |
| | MIA | 46.8 (84.0/41.1/15.2) |
| | PAO | 64.3 (91.6/51.4/49.7) |
| | PIT | 59.9 (82.0/53.2/44.4) |
| | WDC | 39.9 (80.1/39.7/0.0) |
| KITTI | Germany | 66.8 (85.2/58.1/57.2) |
| KITTI-360 | Germany | 43.3 (76.3/22.3/31.4) |
| Lyft | Palo Alto | 60.5 (87.4/45.7/48.5) |
| nuScenes | boston-seaport | 61.4 (75.4/51.7/57.1) |
| | singapore-hollandvillage | 30.3 (67.4/23.4/0.1) |
| | singapore-onenorth | 50.5 (66.8/53.0/31.8) |
| | singapore-queenstown | 51.6 (68.5/58.9/27.5) |
| Waymo | other | 44.9 (72.4/44.8/17.6) |
| | phx | 63.3 (87.0/55.7/47.1) |
| | sf | 65.1 (83.7/58.8/52.8) |

Table 13: Surrounding view 3D detection results: ablation study on sensor alignment approaches. All models are trained on Argoverse2, nuScenes and Waymo.

| K-sync | E-aware | Ego | A | N | W | L | K | K360 | avg. |
|--------|---------|-----|------|------|------|------|------|------|------|
| | | | 48.0 | 40.4 | 54.8 | 0.6 | 6.2 | 0.7 | 25.1 |
| | | ✓ | 48.6 | 41.0 | 53.8 | 0.0 | 3.6 | 0.0 | 24.5 |
| | ✓ | | 49.5 | 39.7 | 54.7 | 1.8 | 7.4 | 1.8 | 25.8 |
| | ✓ | ✓ | 47.4 | 40.8 | 53.3 | 0.2 | 4.4 | 0.6 | 24.4 |
| ✓ | | | 52.2 | 46.5 | 55.2 | 22.2 | 22.0 | 11.0 | 34.9 |
| ✓ | | ✓ | 50.1 | 47.2 | 54.2 | 30.1 | 36.3 | 22.1 | 40.0 |
| ✓ | ✓ | | 52.0 | 46.7 | 55.5 | 31.1 | 26.2 | 15.8 | 37.9 |
| ✓ | ✓ | ✓ | 52.1 | 47.5 | 54.8 | 31.4 | 39.7 | 24.0 | 41.6 |

Table 14: Surrounding view 3D detection results: models are trained on different combinations of Waymo and nuScenes.

| test/ train(W+N) | 0.00+1.00 | 0.01+0.99 | 0.10+0.90 | 0.33+0.67 | 0.5+0.5 | 0.67+0.33 | 1.00+0.00 |
|------------------|-----------|-----------|-----------|-----------|---------|-----------|-----------|
| A | 1.0 | 6.4 | 9.1 | 13.0 | 14.5 | 16.0 | 4.3 |
| N | 32.5 | 32.5 | 32.3 | 32.1 | 31.5 | 30.4 | 0.2 |
| W | 0.3 | 23.4 | 36.7 | 45.8 | 45.6 | 47.0 | 46.2 |

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

# Derivation of Eq.1 and Eq. 3 in the main paper

In this page, we will derive Eq. 1 and Eq. 3 from the main paper step by step. We will first analyze Eq. 3 since Eq. 1 can be easily derived from Eq. 3

## Derivation of Eq. 3

Given an origin at a fixed location of our ego car, with the x-axis along the direction of travel and the z-axis perpendicular to the ground pointing upwards, we define a right-handed Euclidean coordinate system as the ego frame. Assume we have a forward-facing camera along the x-axis, centered at $(t_x, t_y, t_z)$. According to the pinhole camera model, we have camera intrinsic matrix $\mathbf{K}$ and extrinsic matrix $[\mathbf{R}|\mathbf{t}]$, where $\mathbf{R}$ is a $3 \times 3$ rotation matrix. Now $\mathbf{R}, \mathbf{K}, \mathbf{t}$ can be represented as follows:

$$\mathbf{K} = \begin{bmatrix} f_x & 0 & c_x \\ 0 & f_y & c_y \\ 0 & 0 & 1 \end{bmatrix}$$

$$\mathbf{R} = \begin{bmatrix} 0 & -1 & 0 \\ 0 & 0 & -1 \\ 1 & 0 & 0 \end{bmatrix}$$

$$\mathbf{t} = \begin{bmatrix} t_y \\ t_z \\ -t_x \end{bmatrix},$$

where $f_x$, $f_y$ are the focal lengths along the x- and y-axis of the camera coordinate system, and $c_x$, $c_y$ are the pixel offsets on the image. For simplicity we assume $f_x = f_y$ and denote them as $f$ so:

$$\mathbf{K} = \begin{bmatrix} f & 0 & c_x \\ 0 & f & c_y \\ 0 & 0 & 1 \end{bmatrix}$$

Now for a point $\mathbf{p}_0(x_0, y_0, z_0)$ in the ego frame, we want to find the relationship between $\mathbf{p}_0$ and its projection on the image plane. Let's assume its image position as $(u, v)$.

For the first step, we transform $\mathbf{p}_0$ into the camera coordiate system:

$$\mathbf{p}_{cam} = \mathbf{R} \times \mathbf{p}_0 + \mathbf{t} = \begin{bmatrix} 0 & -1 & 0 \\ 0 & 0 & -1 \\ 1 & 0 & 0 \end{bmatrix} \times \begin{bmatrix} x_0 \\ y_0 \\ z_0 \end{bmatrix} + \begin{bmatrix} t_y \\ t_z \\ -t_x \end{bmatrix} = \begin{bmatrix} -y_0 + t_y \\ -z_0 + t_z \\ x_0 - t_x \end{bmatrix}$$

Here, we assign $d = x_0 - t_x$ as the 'depth' of $\mathbf{p}_0$ in the camera, and use $X = -y_0 + t_y$, $Y = -z_0 + t_z$ as shorthand.

For the second step, we transform $\mathbf{p}_{cam}$ to the image coordinate system:

$$\mathbf{p}_{img} = \mathbf{K} \times \mathbf{p}_{cam} = \begin{bmatrix} f & 0 & c_x \\ 0 & f & c_y \\ 0 & 0 & 1 \end{bmatrix} + \begin{bmatrix} X \\ Y \\ d \end{bmatrix} = \begin{bmatrix} fX + c_x d \\ fY + c_y d \\ d \end{bmatrix}$$

For the last step, we collapse depth and project $\mathbf{p}_{img}$ onto the imaging plane to obtain the values of $(u, v)$:

$$\mathbf{p}_{2D} = \mathbf{p}_{img}/d = \begin{bmatrix} \frac{fX}{d} + c_x \\ \frac{fY}{d} + c_y \\ 1 \end{bmatrix} = \begin{bmatrix} u \\ v \\ 1 \end{bmatrix}$$

In summary:

$$\mathbf{p}_{img} = d(u, v, 1)^T$$

$$\mathbf{p}_{img} = \mathbf{K} \times \mathbf{p}_{cam} = \mathbf{K} \times (\mathbf{R} \times \mathbf{p}_0 + \mathbf{t})$$

So:

$$d(u, v, 1)^T = \mathbf{K}(\mathbf{R}\mathbf{p}_0 + \mathbf{t})$$

Assuming that the points are represented in homogeneous coordinates, then $\mathbf{p}_0 = (x_0, y_0, z_0, 1)^T$, and we can simplify the relationship:

$$d(u, v, 1)^T = \mathbf{K}(\mathbf{R}\mathbf{p}_0 + \mathbf{t}) = \mathbf{K}(\begin{bmatrix} \mathbf{R} & \mathbf{t} \\ 0 & 1 \end{bmatrix} \mathbf{p}_0) = \mathbf{K}\mathbf{T}\mathbf{p}_0$$

Now:

$$d(u, v, 1)^T = \mathbf{KTp}_0,$$

where

$$\mathbf{T} = \begin{bmatrix} \mathbf{R} & \mathbf{t} \\ 0 & 1 \end{bmatrix}, d = x_0 - t_x$$

## Derivation of Eq. 1

With the derivation and notations of Eq. 3, we can directly derive Eq. 1. Assume that a 3D box (e.g., a car) is centered at $\mathbf{p}_0$, and the origin of ego frame is on the ground, then $2z_0$ is the height dimension of the box. We project $\mathbf{p}_1 = (x_0, y_0, 0)$ and $\mathbf{p}_2 = (x_0, y_0, 2z_0)$ onto the image to get the upper and lower bounds of the box, and the difference between them should be the pixel height of this box. From the derivation of Eq. 3, we know:

$$v = \frac{fY}{d} = \frac{f(-z_0 + t_z)}{d}$$

Similarly, for $\mathbf{p}_1$ and $\mathbf{p}_2$ we have:

$$v_1 = \frac{ft_z}{d} + c_y,$$

$$v_2 = \frac{f(-2z_0 + t_z)}{d} + c_y$$

So the pixel height of the box would be:

$$height_{2D} = |v_1 - v_2| = f \times \frac{2z_0}{d} = f \times \frac{2z_0}{x_0 - t_x}$$

Let the 3D size $S = 2z_0$, and the 2D pixel size $s_{pixel} = height_{2D}$, then we have the Eq. 1 in main paper:

$$s_{pixel} = f \times \frac{S}{x_0 - t_x}$$