# OpenReview forum: "Cross-Dataset Sensor Alignment: Making Visual 3D Object Detector Generalizable"
_robot-learning.org/CoRL/2023/Conference — CoRL 2023 Poster_

### Official Review · Reviewer_NKfY · 2023-06-29

**Confidence:** 3
**Originality:** Good
**Technical Quality:** Good
**Clarity Of Presentation:** Fair
**Impact:** 4

**Recommendation:**

Weak Reject: I recommend rejecting the paper, but will not argue for my recommendation if the majority of other reviewers have a different opinion.

**Review:**

**STRENGTHS**

* The paper's contributions result in **very significant (game-changing) cross-dataset generalization results in the realm of 3d detection for AV datasets**.

* The above is **achieved with quite simple data preprocessing tricks** -- nice!

* **The contribution of each of the three different alignment techniques are well-motivated** (see Section 3.4) and empirically (see e.g. Table 3, Table 4, and Table 4 in the SM).

* Good that so **many (6) datasets are used** and evaluated on.

* Good that **more than one detection algorithm is tested** (both DETR3D and BEVDet).

**WEAKNESSES**

* The paper focuses on 3d object detection for AV datasets. **I feel however that the text is lacking some discussion around the same sensor alignment methodology on other vision tasks for AV datasets** -- do the authors hypothesize that similar sensor alignments are helpful there too? Why/why not? Have others done that already perhaps?

* **Quite a few things were not clear to me and/or some things were structurally a bit off:**
    - I didn't understand why the mAP was the same for Argo2 and Waymo in Fig. 2 (?).
    - Line 62: "... over 20 mAP on most datasets". Line 70: "... more than 25 mAP on all datasets" <-- these statements are somewhat inconsistent.
    - Line 97: "... avoid employing data augmentation ... " <-- this is unclear to me.. why would one skip that? In particular, the data augmentation might have helped the other approaches that do not apply the various alignment techniques (**making the overall comparisons less fair (?)**).
    - Line 118: Unclear to me what MMDetection3D is. Please add some reference in the paper.
    - Line 136: "... thresholds at 05, 0.3 and 0.3 ... " <-- I guess one of these is a typo (?).
    - Some tables occur quite far from where they are mentioned and in "reverse" order (so Table X may be higher up in the paper than Table Y, even though X was introduced after Y). Table 3 is for example quite far away.
    - Speaking of tables, the various terminologies in the table are not introduced prior to introducing the table (e.g. "Direct"). Please add all abbreviations and explain all properties / settings clearly before a table gets introduced.
    - There are quite a few numbers / tables (good!) but it can get hard to grasp everything clearly, so I would recommend adding another column besides "avg." which is something like "avg-not-same" i.e. which takes the average only for those DIFFERENT datasets (thus avg-not-same explicitly averages the scores for cross-dataset generalization, by not including the same dataset on which it was trained). It could also be good to highlight the best entries in each table with bold or similar.
    - Why is "Sync Intrinsic" for N different in Table 4 compared to Table 3? (they are the same for "Direct" N and "Sync Extrinsic and Ego center").
    - Line 312: '... lack of autonomous driving datasets with highly diverse camera poses ... ' <-- **is that true, why would there be so few camera poses?**

* **Some things are a bit lacking regarding experimental evaluations, e.g:**
    - I understood why one had to take the intersection among all classes to get a universal method. However, **it would also have been good to add some results which include more than just those 3 categories** (that would imply that some of the 6 datasets would have to be omitted). The models could then anyway be evaluated on the omitted datasets (albeit not for all categories, as some are missing from those datasets).
    - **It was not clear to me whether "Sync Extrinsic and Ego center" also includes intrinsic synchronization (?)** If not, I recommend showing results for when using all three things at the same time.
    - **For the other BEVDet detector (SM), why did you only use the "Direct" approach for its evaluations, i.e. why did you not showcase your sensor adjustment strategy here?**

* The paper seems **quite hastily written with quite a few typos** across the paper (and SM), such as:
    - Line 88: No space between "domain" and "[20, 21, 22, 23, 24, 25, 26, 27, 28]" (this happens in many other places too).
    - Line 88: "Unlike theses methods are trained on ..." <-- "Unlike these methods WHICH are trained on ..., our research ..." (skip 'whereas' here too).
    - Line 99: "... are several works have ... < "... are several works THAT have ..."
    - Line 104: There is too little vertical space between this line and Table 1.
    - Line 169: "... the data diversity, We ..." <-- should be small 'w' on 'we'.
    - Line 192: "Fig. 1 and Fig. 1" <-- duplicate
    - Etc etc <-- please go over the paper in detail to fix all typos etc.

**Quality Of The Limitations Section:**

Limitations are addressed clearly

**Questions For Rebuttal:**

Please see my bolded questions / statements under "WEAKNESSES" and prioritize those. Then try to address any other comments that include question-marks, and or things that should be clarified.

**Robotics Focus:**

Highly relevant to robotics but no hardware experiments

**Summary Of Paper:**

This work studies the generalization of camera-based 3D object detection models across different datasets for autonomous driving. It shows that existing models are highly sensitive to variations in sensor configuration, and proposes a method to align different sensor settings using camera intrinsic and extrinsic corrections, as well as ego frame alignment. These alignment steps are shown to be crucial for cross-dataset generalization (simply merging all datasets as they are and training yields much worse results) -- overall, the results include 6 common AV object detection datasets and two 3D object detection models, and improvements are shown for all.

**Summary Of Recommendation:**

As mentioned, really great empirical results across a wide range of AV datasets! The paper requires quite a bit of polishing / clarifications prior to acceptance, however (see "WEAKNESSES"). Weak reject for now, then let's see during the rebuttal.

---

> ### Comment · Reviewer_NKfY · 2023-08-15
> **Updated recommendation: Weak accept**
>
> Please refer to my comment to the rebuttal:
>
> """
> This post is just to acknowledge that I've carefully read the rebuttal (both the rebuttal to my own review, but also that of the other reviewers). I believe the rebuttals were satisfactory overall, and I feel the authors have put lots of efforts into the various polishes that I requested (recall my prior recommendation: "As mentioned, really great empirical results across a wide range of AV datasets! The paper requires quite a bit of polishing / clarifications prior to acceptance, however (see "WEAKNESSES"). Weak reject for now, then let's see during the rebuttal.").
>
> Given this, I feel the work is now at the level of 'weak accept' (instead of 'weak reject' as prior) and I have updated my rating to this.
> """

---

### Official Review · Reviewer_JvHZ · 2023-07-12

**Confidence:** 3
**Originality:** Fair
**Technical Quality:** Good
**Clarity Of Presentation:** Very Good
**Impact:** 3

**Recommendation:**

Weak Reject: I recommend rejecting the paper, but will not argue for my recommendation if the majority of other reviewers have a different opinion.

**Review:**

Pro:

  1. The authors conducted comprehensive experiments to show that simply combining multiple datasets cannot improve the model’s generalization capability if the datasets are collected with different setups (with different data distributions).
  2. This paper is well-written, well-structured, and easy to follow.

Cons:

  1. The performance drop due to different camera intrinsic/focal lengths between training and testing datasets is trivial to me. Resizing images from their original focal length to a predefined focal length sounds reasonable but cannot address the spatial misalignment among the datasets during collection (ratio/scale of the object in the image) and sounds trivial.

  2. Sec 4.2 regarding the effect of camera extrinsic is unclear to me, in order to get the recipient field in Fig.3(b), a real-world size S of the object is required, which is not possible in real-world given various types of cars. Furthermore, I believe combining multiple datasets with wide extrinsic range could alleviate this problem.

  3. The Intrinsic Synchronization proposed by the authors requires the camera intrinsic of each image/dataset to be given as prior knowledge, which somehow still limits the generalization in the wild.

  4. The authors proposed three modules, namely intrinsic, extrinsic, and ego frame alignment, which to me are trying to do the same thing, i.e., shifting multiple datasets to the same distribution. Especially the ego frame alignment is achieved by grid search and seems specifically useful to the query-based method like DETR3D, which also limits the generalization. For example, I think a pure image-based detector with large-scale multiple datasets for training can simply deal with this problem.

Minor:

  1. I believe there’s a typo in L. 136: ….thresholds at 0.5, 0.3 and 0.3…

  2. L. 138,  average the -> the average



**Quality Of The Limitations Section:**

Additional details required

**Questions For Rebuttal:**

Questions:

  In Sec 4.2, could the authors explain why the image patch and p_0 are invariant to t_x according to the equation in L.258?

**Robotics Focus:**

Highly relevant to robotics but no hardware experiments

**Summary Of Paper:**

In this work, the authors investigate the generalization of visual 3D object detection in autonomous driving across datasets by aligning the sensor configuration. The authors claim that simply aligning the sensor configuration including camera intrinsic, camera extrinsic, and ego coordinate system can significantly mitigate the performance gap between different datasets.

**Summary Of Recommendation:**

Overall, I think the paper is well-written and well-structured. However, the problem and method proposed by the authors are trivial to me and very specific to the chosen baseline. There is not much novelty w.r.t. the algorithm. Therefore, I would lean toward a weak rejection.

---

### Official Review · Reviewer_iJm9 · 2023-07-19

**Confidence:** 4
**Originality:** Very Good
**Technical Quality:** Very Good
**Clarity Of Presentation:** Very Good
**Impact:** 2

**Recommendation:**

Weak Accept: I recommend accepting the paper, but will not argue for my recommendation if the majority of other reviewers have a different opinion.

**Review:**

## Strength
1. Interesting and important task: make the camera-based 3D detector generalizable.
2. Extensive experiments and convincing results.
3. Clear writing (but a bit wordy, especially in the table/fig)

## Weakness
1. The baseline performance is too trival and a bit lack practicality. I think no one will directly use a detector trained on different sensor configurations.
A reasonable setting for monocular 3D detection is to detect the object with respect to the camera origin, and then transform to vehicle frame by applying the extrinsics.
For example, a monocular detector could be designed in this way: first detect 2D bbox, then predict depth, rotation and box3d size, and then transform the vehicle frame. We can applied the intrinsics and extrinsics in the post-processing, so I think it can be generalizable to different dataset.
2. No experiments on boosting the performance with extra data. e.g. train with 5+1 dataset and test on the last one,and see if the performance can be improved with more trained dataset.



**Quality Of The Limitations Section:**

Limitations are addressed clearly

**Questions For Rebuttal:**

Please address my comments on the weakness section.

**Robotics Focus:**

Highly relevant to robotics but no hardware experiments

**Summary Of Paper:**

This work focus on generalizable camera-only 3D detector for self-driving: train on one dataset and test on the others.
Specifically, the challenges exist in intrinsic, extrinsic and ego-coordinate system difference.
This work propose some simple yet effective solution to address these issues: resize image to match the intrinsic,  Extrinsic Aware Module  to handle the extrinsic and ego-frame alignment to handle different ego height..
Extensive experiments are conducted on 6 datasets to validate the effectiveness.

**Summary Of Recommendation:**

Overall, this work studies an interesting task. But the baselines are a bit too weak, the solution is somewhat trivial but do shows improvement.
I gave an weak accept because it's an interesting task/setting that no other work has addressed (but my reading is limited).

---

### Official Review · Reviewer_vndD · 2023-07-24

**Confidence:** 4
**Originality:** Excellent
**Technical Quality:** Very Good
**Clarity Of Presentation:** Excellent
**Impact:** 4

**Recommendation:**

Strong Accept: I recommend accepting the paper and will argue for my recommendation even if other reviewers hold a different opinion.

**Review:**

Overall, this paper is well-motivated, very well-written, and easy to follow. The visualizations made it easier to understand the problems and solutions. Lastly, the results are impressive.

There are a few bits in the main paper that were a bit difficult for me to understand. I would appreciate it if the authors can address them in the rebuttal.

1. In the abstract, line 5, the authors mentioned about weather conditions being the same between datasets can be a cause of overfitting. However, there is no mention of this issue in the main paper. Even the augmentations are dropped (line 129). I would like to know why.

2. Adding more datasets results in more data for each of the categories. However, it can also exacerbate the data-imbalance problem. For example, adding KITTI-360 that has only vehicles but no cyclists or pedestrians. I would like to know if the authors looked at the data statistics and addressed this problem during training.

3. I could not understand the derivation of Eq. 1 and Eq. 3. I would appreciate it if the authors can explain how those equations are extracted from the pinhole camera model. Also, please give the dimensions of each variable (S, s_pixel, and p0) used in them.

4. In lines 287-288, I could not understand how the optimal ego settings are selected based on Figure 4. Please explain it a bit more.

5. Lastly in the limitations section, the authors mention that they could not study the impact of camera rotation. By this do they mean most of the data in the datasets is collected on a straight road? If yes, can the evaluation be done on the isolated sequences where the vehicle is turning?

**Quality Of The Limitations Section:**

Additional details required

**Questions For Rebuttal:**

Please see the above comments

**Robotics Focus:**

Highly relevant to robotics but no hardware experiments

**Summary Of Paper:**

This paper proposes a method to make 3D object detectors generalize by focusing on cross-dataset sensor-setup alignment. This paper argues that differences in camera intrinsics and extrinsic, and ego-frame alignment, also attributed as "sensor misalignment" plays a significant role in the generalization capability of a detector. To support this extensive empirical studies were conducted on 6 different datasets. The proposed simple sensor alignment approaches with intrinsic Synchronization, EAM module, and ego-frame alignment are able to address the aforementioned problems. Despite their simplicity, they are proven to work very well in practice and improve the mAP metric substantially.

**Summary Of Recommendation:**

3D object detection is a very relevant and important problem in the field of robotics. This paper explains the fundamental problems of why just adding new data can not improve the performance of a detector and did a very good job of addressing those problems with simple yet effective solutions. These kinds of findings (both theoretical and empirical) are very crucial for the advancement of research in this direction.

---

### Author Response · Authors · 2023-08-14
**General Response**

We would like to express our sincere gratitude for the valuable feedback provided for our paper. Your suggestions have greatly contributed to enhancing the quality of the work, and we appreciate your interest in understanding these aspects of our work. In response, we have provided a comprehensive discussion addressing these points in our general response, and we kindly invite you to refer to it for detailed information on these topics.

In addition to addressing your concerns, we have corrected typos and clarified expressions as suggested. The revised parts are marked in red, and our **updated paper and supplementary materials** are included in the attachments for your reference.
Thank you once again for your insightful comments.

---

### Decision · Program_Chairs · 2023-08-30

**Decision:**

Accept (Poster)

**Comment:**

The paper investigates the generalization of camera-based 3D object detectors across different datasets for autonomous driving – showing that existing models are highly sensitive to variations in sensor configuration, and proposes a method to align different sensor settings using camera intrinsic and extrinsic corrections, and ego frame alignment. Experiments show that the proposed method for cross-dataset transfer improves over simply merging all datasets as they are, with results on 6 AV object detection datasets and two 3D object detection models.

Reviewers agree that paper is interesting (iJm9, NKfY), well-motivated (vndD, iJm9), easy to follow (vndD, JvHZ), with convincing results across datasets (vndD, iJm9, NKfY). For the rebuttal, reviewers requested additional experiment clarifications and statistics (vndD, iJm9, JvHZ, NKfY) as well as additional experiments with sensor adjustment on BEVDet (NKfY). Post-rebuttal, the ratings have improved to strong accept, weak accept, weak reject, weak accept. There are still a few outstanding concerns on novelty, and the constraints on the design of the base detector (JvHZ) – in particular, the writing may benefit from including a brief discussion on the latter.

Overall, while the paper does not introduce a new algorithm, it shows that tracing the problem back down it roots (i.e. the data) may yield effective and simple solutions. Cross-dataset transfer is also a timely topic in the community, and this work should encourage data retrospection that extends beyond the AV space. I believe the paper to be an excellent reminder of the value behind a first-principles approach, and presents a refreshing take on how "fixing existing data" can be just as (if not more) fruitful at time when "scaling up / more data" remains all the rage.